# Alleviating "Posterior Collapse" in Deep Topic Models via Policy Gradient

**Yewen Li**[1][*]   **Chaojie Wang**[1][*][†]   **Zhibin Duan**[2]
**Dongsheng Wang**[2]   **Bo Chen**[2]   **Bo An**[1]   **Mingyuan Zhou**[3]
[1]Nanyang Technological University   [2]Xidian University   [3]The University of Texas at Austin

## Abstract

Deep topic models have been proven as a promising way to extract hierarchical latent representations from documents represented as high-dimensional bag-of-words vectors. However, the representation capability of existing deep topic models is still limited by the phenomenon of "*posterior collapse*", which has been widely criticized in deep generative models, resulting in the higher-level latent representations exhibiting similar or meaningless patterns. To this end, in this paper, we first develop a novel deep-coupling generative process for existing deep topic models, which incorporates skip connections into the generation of documents, enforcing strong links between the document and its multi-layer latent representations. After that, utilizing data augmentation techniques, we reformulate the deep-coupling generative process as a Markov decision process and develop a corresponding Policy Gradient (PG) based training algorithm, which can further alleviate the information reduction at higher layers. Extensive experiments demonstrate that our developed methods can effectively alleviate "*posterior collapse*" in deep topic models, contributing to providing higher-quality latent document representations.

## 1   Introduction

Topic modeling has become a successful technique for text analysis and been widely applied to various problems in machine learning (ML) [1, 2, 3] and natural language processing (NLP) [4, 5] over the past two decades. Representing documents as bag-of-words (BoW) vectors, vanilla probabilistic topic models (PTMs), with latent Dirichlet allocation (LDA) [6] being the best known representative, typically formulate each document as a mixture over latent topics, where each topic is characterized by a distribution over the terms of the vocabulary and describes an interpretable semantic concept. Although being widely used, the modeling capability of these shallow topic models is still restricted by their single-layer structure, and has difficulty in exploring hierarchical thematic structures. To this end, a series of deep topic models [7, 8, 9] have been developed to extract multi-layer document representations from a text corpus, providing a more intuitive way for users to understand text data.

Recently, benefiting from the development of deep neural networks (DNNs), there has been an emerging research interest to develop neural topic models (NTMs) to boost the performance, efficiency, and usability of topic modeling with DNNs. Specifically, following the framework of variational autoencoder (VAE) [10], most NTMs [11, 12, 13] construct a variational inference network (encoder) to project each document into its stochastic latent representation, and then reconstruct the corresponding BoW observation with a stochastic/deterministic decoder. By modeling the inference/generative process with DNNs, these NTMs are more flexible and scalable than traditional Bayesian PTMs, contributing to performing large-scale downstream tasks, especially in NLP tasks [14, 15].

---

*Equal contributions.
†Corresponding to: Chaojie Wang <chaojie.wang@ntu.edu.sg>.

36th Conference on Neural Information Processing Systems (NeurIPS 2022).

"*Posterior collapse*" has been widely criticized in the field of generative model [16, 17], and the occurrence of this phenomenon will cause the approximated posterior $q_\phi(z|x)$ collapses to it non-information prior distribution $p_\theta(z)$, leading their KL divergence to be close to zero [10, 16, 17, 18]. For deep topic models, despite achieving attractive performance, existing PTMs or NTMs still suffer from different degrees of "*posterior collapse*", which causes their exhibiting similar or meaningless patterns at higher layers [19, 20, 21]. Although there have been several deep NTMs [20, 21] trying to alleviate this issue by constructing more flexible inference networks, the collapse phenomenon in deep NTMs may not be solved in essence, because the true posterior provided by the generative model and the objective function for optimization remain almost unchanged [18].

To extract higher-quality hierarchical latent document representations, in this paper, we develop a deep-coupling generative process equipped with a Policy Gradients (PG) based training algorithm for existing deep topic models. The main contributions of this work are as follows:

- We develop a deep-coupling generative process for deep topic models, which incorporates skip connections into the generation of documents to alleviate "*posterior collapse*".
- We take a specific NTM as an example to explain how to construct a deep topic model with the deep coupling generation process, and develop a deep-coupling hierarchical Embedding Topic Model ($dc$-ETM), which can be extended to other deep topic models.
- Utilizing the property of sequence-like generation process, we design a PG-based training algorithm for $dc$-ETM, which can further alleviate the information reduction at higher layers.
- Compared to existing deep topic models, extensive experimental results show that $dc$-ETMs can lead to less "*posterior collapse*" and provide higher-quality latent representations.

## 2 Related Work

**Probabilistic Topic Model:** Deep PTMs [7, 8, 9, 22] are developed to infer multi-layer document representations, whose adjacent layers are connected with specific factorization. For instance, gamma belief network (GBN) [8] is constructed via factorizing the shape parameters of the gamma distributed latent representations; DPFA [7] extends PFA [23] into a multi-layer version but is restricted to model binary topic usage patterns; DirBN [9] is developed via factorizing the Dirichlet distributed topic matrix. Although providing readily interpretable multi-layer latent document representations, the representation capability of these deep PTMs is limited by adopting CRT distribution to upward propagate data information to higher layers with their backbones [19].

**Neural Topic Model:** Most existing NTMs [12, 20, 21, 24, 25, 26] can be viewed as extensions of PTMs under the VAE framework and focus on modeling the generative/inference process with DNNs. For instance, one popular research direction of NTMs is to develop more flexible inference network with reparametrization tricks [12, 20] and the other could be incorporating word embeddings into the generative model [21, 24]. However, as far as we know, few efforts have been made to alleviate the phenomenon of "*posterior collapse*" in NTMs by modifying its generative process, which is a great challenge under the framework of topic modeling and also the main contribution of this work.

Besides, distinct from the way of combining reinforcement learning (RL) with topic models in previous works [27, 28, 29], our work is the first to formulate the topic modeling generative process as a sequential decision making one to incorporate RL-based training algorithms, which focuses on providing higher-quality latent document representations by alleviating "*posterior collapse*".

## 3 Deep-Coupling Generative Process for Deep Topic Models

To give an intuitive insight on "*posterior collapse*" in deep topic models, we provide a detailed introduction for "*posterior collapse*" in Appendix J and then visualize the higher-level topics learned by a recent popular NTM named SawETM [21] in Fig. 3, which exhibit similar semantic patterns and limit its representation capability. Then, we take SawETM as an example, but not limited to this, to illustrate how to construct a deep topic model with the deep coupling generation process, leading to a novel $dc$-ETM in Fig. 1(c). Compared to the usual structures of deep PTMs and NTMs shown in Fig. 1(a) and 1(b), besides the design of inference network, the main difference of $dc$-ETM is incorporating skip connections into the generation of documents, enforcing strong links between the document and its multi-layer latent representations to alleviate "*posterior collapse*".

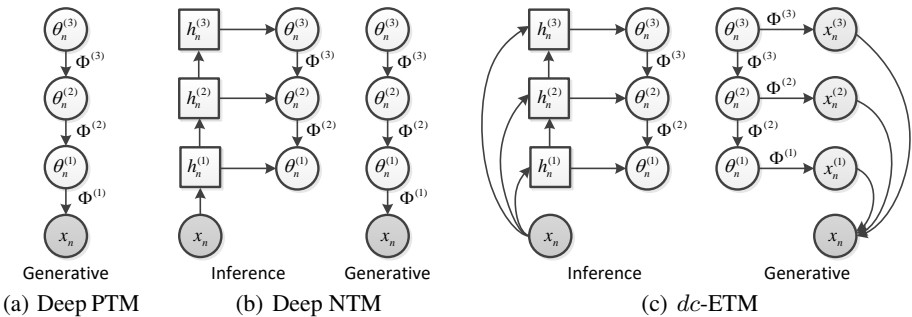

<table>
</table>

|  (a) Deep PTM | (b) Deep NTM | (c) $dc$-ETM |

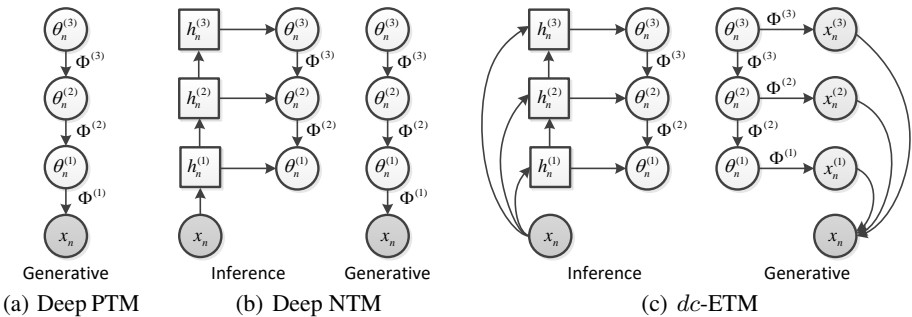

(a) Deep PTM    (b) Deep NTM    (c) $dc$-ETM

Figure 1: The overview of the network structure of (a) deep PTM, (b) deep NTM, and (c) $dc$-ETM developed in this paper, where the symbol definitions are consistent with those in Sec. 3.1.

## 3.1 Deep-Coupling Hierarchical Embedding Topic Model

As a usual VAE-like model, the developed $dc$-ETM consists of a generative model (decoder) and an inference network (encoder). Below, we focus on presenting the generative model of $dc$-ETM, which can be flexibly applied for other deep topic models to alleviate "*posterior collapse*", and leave the details of the inference network to Appendix A.

**Generative Model:** Given a text corpus consisting of $N$ documents $\boldsymbol{X} = \{\boldsymbol{x}_n\}_{n=1}^N$, each document can be represented as a high-dimensional sparse BoW vector $\boldsymbol{x}_n \in \mathbb{Z}^{K^{(0)}}$, where $\mathbb{Z} = \{0, 1, ...\}$ and $K^{(0)}$ denotes the vocabulary size. Then, from top to bottom, the generative model of the $dc$-ETM with $L$ hidden layers can be formulated as

$$\boldsymbol{\theta}_n^{(l)} \sim \text{Gam}(\boldsymbol{\Phi}^{(l+1)}\boldsymbol{\theta}_n^{(l+1)}, 1/c_n^{(l+1)}), l = 1, ..., L-1, \cdots, \boldsymbol{\theta}_n^{(L)} \sim \text{Gam}(\boldsymbol{r}, 1/c_n^{(L+1)}), \tag{1}$$

$$\boldsymbol{x}_n \sim \text{Pois}(\sum_{l=1}^L \alpha^{(l)}\hat{\boldsymbol{\Phi}}^{(l)}\boldsymbol{\theta}_n^{(l)}), \ \boldsymbol{\alpha} = \text{Softmax}(\boldsymbol{\xi}), \ \boldsymbol{\phi}_k^{(l)} = \text{Softmax}(\boldsymbol{\beta}^{(l-1)^T}\boldsymbol{\beta}_k^{(l)}), l = 1, ..., L-1,$$

where, $\boldsymbol{\Phi}^{(l)} \in \mathbb{R}_+^{K^{(l-1)} \times K^{(l)}}$ denotes the topic matrix (factor loading) and each column $\boldsymbol{\phi}_k^{(l)} \in \mathbb{R}_+^{K^{(l-1)}}$ indicates a specific topic (factor) at layer $l$; $\boldsymbol{\theta}_n^{(l)} \in \mathbb{R}_+^{K^{(l)}}$ denotes the gamma distributed latent representation (topic proportions) at layer $l$, $K^{(l)}$ denotes the number of hidden units (topics) at layer $l$. Under the Poisson likelihood, the observed multivariate count vector $\boldsymbol{x}_n$ is first factorized into $L$ equal-size latent matrix $\{\alpha^{(l)}\hat{\boldsymbol{\Phi}}^{(l)}\boldsymbol{\theta}_n^{(l)}\}_{l=1}^L$, where, $\hat{\boldsymbol{\Phi}}^{(l)} \in \mathbb{R}_+^{K^{(0)} \times K^{(l)}}$ can be regarded as the projection of topic matrix $\boldsymbol{\Phi}^{(l)}$ to the observation space and the detailed definition will be discussed in the next paragraph; $\alpha^{(l)}$ denotes the importance weight of $\hat{\boldsymbol{\Phi}}^{(l)}\boldsymbol{\theta}_n^{(l)}$ for generating the observation $\boldsymbol{x}_n$, and the summation of the whole weight vector $\boldsymbol{\alpha} \in \mathbb{R}_+^L$ is constrained to be equal to one with a Softmax normalization. Then, the latent representation $\boldsymbol{\theta}_n^{(l)}$ at layer $l$ is further factorized into the product of the topic matrix $\boldsymbol{\Phi}^{(l+1)} \in \mathbb{R}_+^{K^{(l)} \times K^{(l+1)}}$ and topic proportions $\boldsymbol{\theta}_n^{(l+1)} \in \mathbb{R}_+^{K^{(l+1)}}$ at the next layer under the shape of gamma distribution. The top layer's latent representation $\boldsymbol{\theta}_n^{(L)}$ shares the same gamma shape parameters $\boldsymbol{r} \in \mathbb{R}_+^{K^{(L)}}$ and we apply a gamma distributed prior on the scale parameters $c_n^{(l)}$ for $l \in \{2, ..., L+1\}$. With the recent popular distributed topic representation in NTMs [24, 30], each topic $\boldsymbol{\phi}_k^{(l)}$ is treated as the result of applying a Softmax normalization on the inner product of its distributed representation $\boldsymbol{\beta}_k^{(l)} \in \mathbb{R}^D$ and topic embedding matrix $\boldsymbol{\beta}^{(l-1)} \in \mathbb{R}^{D \times K^{(l-1)}}$ at the previous layer, where $D$ denotes the dimension of the embedding space.

The projections of topic matrices to the observation space, denoted as $\{\hat{\boldsymbol{\Phi}}^{(l)}\}_{l=1}^L$, build the straightforward connections between the document $\boldsymbol{x}_n$ and its multi-layer latent representations $\{\boldsymbol{\theta}_n^{(l)}\}_{l=1}^L$, which alleviates the information reduction at higher layers by sharing the pressure of document modeling with all hidden layers. To reduce the computation and storage cost of the developed $dc$-ETM, we develop two variants for $\hat{\boldsymbol{\phi}}_k^{(l)} \in \mathbb{R}_+^{K^{(0)}}$ without introducing any extra parameter. The one variant is adopting the property of topic hierarchy elaborated in Sec. 3.2 to obtain each $\hat{\boldsymbol{\phi}}_k^{(l)}$ by

successively multiplying topic matrices at lower layers as

$$\hat{\phi}_k^{(l)} = \prod_{t=1}^{l-1} \mathbf{\Phi}^{(t)} \phi_k^{(l)},$$ (2)

and the other variant is treating the projection $\hat{\phi}_k^{(l)}$ as the result of the inner product of its distributed representation $\boldsymbol{\beta}_k^{(l)}$ and the word embedding matrix $\boldsymbol{\beta}^{(0)}$ at the observed space, as follows

$$\hat{\phi}_k^{(l)} = \text{Softmax}(\boldsymbol{\beta}^{(0)^T} \boldsymbol{\beta}_k^{(l)}).$$ (3)

We emphasize that the first variant can be used to extend most existing deep topic models, while the latter is limited to NTMs equipped with topic embedding techniques. We use the suffix $-\alpha$ and $-\beta$ to distinguish the variants defined in Eq. (2) and Eq. (3), and their detailed implementations can be found in Appendix I.

Generally speaking, the deep-coupling generative process in $dc$-ETM not only preserves the hierarchy of traditional deep topic models, leading to multi-layer document representations to enhance the modeling capability and interpretability, but also alleviates the issue that the amount of information will decrease rapidly with the network going deeper, benefiting from building the straightforward connections between observation $\boldsymbol{x}_n$ and its higher-level latent representations $\{\boldsymbol{\theta}_n^{(l)}\}_{l>1}$. Besides alleviating "*posterior collapse*", the characteristics of deep-coupling network structure of $dc$-ETM also brings us a new view to design the corresponding inference network and training algorithm.

**Inference Network**: The details of the inference network of $dc$-ETM can be found in Appendix A.

## 3.2 Model Property

**Sequence-like Generative Process:** Taking advantages of the properties of the Poisson distribution, the original generative process of the observed data $\boldsymbol{x}_n$ defined in Eq. (1) can be rewritten as:

$$\boldsymbol{x}_n = \sum_{l=1}^{L} \boldsymbol{x}_n^{(l)}, \ \boldsymbol{x}_n^{(l)} \sim \text{Pois}(\alpha^{(l)} \hat{\mathbf{\Phi}}^{(l)} \boldsymbol{\theta}_n^{(l)}),$$ (4)

where $\boldsymbol{x}_n^{(l)}$ denotes the augmented observation at layer $l$, and is generated from the Poisson distribution with a rate of $\alpha^{(l)} \hat{\mathbf{\Phi}}^{(l)} \boldsymbol{\theta}_n^{(l)}$. Then, the observed data $\boldsymbol{x}_n$ can be regarded as not only the summation over these augmented vectors $\{\boldsymbol{x}_n^{(l)}\}_{l=1}^{L}$, but also equal to the weighted summation over the latent vectors $\{\hat{\mathbf{\Phi}}^{(l)} \boldsymbol{\theta}_n^{(l)}\}_{l=1}^{L}$ on the mean, where the weight vector $\boldsymbol{\alpha}$ satisfies the constraint $\sum_{l=1}^{L} \alpha^{(l)} = 1$.

Rethinking the generative process of the developed $dc$-ETM reformulated in Eq. (4), the set of augmented observation vectors $\{\boldsymbol{x}_n^{(l)}\}_{l=1}^{L}$ can naturally form an observation sequence $[\boldsymbol{x}_n^{(L)}, ..., \boldsymbol{x}_n^{(1)}]$ by sorting these vectors according to their dependencies in the generative process (from deep to shallow). For each hidden layer (time step) $l$, the generative process will first incorporate the prior information passing from deeper hidden layers $\{\boldsymbol{\theta}_n^{(t)}\}_{t>l}$, and then generate the latent representation $\boldsymbol{\theta}_n^{(l)}$ at the current layer (time step), which not only is supposed to generate the current observation vector $\boldsymbol{x}_n^{(l)}$ under the Poisson likelihood, but also introduces the information into the shape parameter of the following gamma distributed latent representation $\boldsymbol{\theta}_n^{(l-1)}$ at the next layer (time step).

Thus, the deep-coupling generative process of $dc$-ETM originally defined in Eq. (1) can be naturally reinterpreted from the perspective of sequence generation, and its reformulation defined in Eq. (4) can be also equivalently reformulated as:

$$\boldsymbol{x}_n \sim \sum_{l=1}^{L} \text{Pois}(\alpha^{(l)} \hat{\mathbf{\Phi}}^{(l)} \boldsymbol{\theta}_n^{(l)}),$$ (5)

providing an intuitive insight for the decomposition of the likelihood function in Sec. 4.1.

**Hierarchical Semantic Topics:** The developed $dc$-ETM can naturally interpret each semantic topic $\phi_k^{(l)}$ at layer $l$ by visualizing its projection to the vocabulary space calculated as

$\{[\prod_{t=1}^{l-1} \boldsymbol{\Phi}^{(t)}] \boldsymbol{\phi}_k^{(l)}\}_{k=1}^{K^{(l)}}$, and each document can also be roughly seen as a random mixture over $K^{(l)}$ topics with $\boldsymbol{\theta}_n^{(l)}$ being the corresponding topic proportions at layer $l$ as

$$\mathbb{E}\left[\boldsymbol{x}_n | \boldsymbol{\theta}_n^{(l)}, \left\{\boldsymbol{\Phi}^{(t)}, c_n^{(t)}\right\}_{t=1}^l\right] = \left[\prod_{t=1}^l \boldsymbol{\Phi}^{(t)}\right] \frac{\boldsymbol{\theta}_n^{(l)}}{\prod_{t=2}^l c_n^{(t)}}, \tag{6}$$

which can be obtained with the law of total expectation. Moreover, similar to the underlying idea of the deep learning, the topics learned by $dc$-ETM tend to be more specific at lower (bottom) layers and those at higher (top) layers are more general, as shown in Fig. 7.

Secondly, in $dc$-ETM, both words $\boldsymbol{\beta}^{(0)} \in \mathbb{R}^{D \times K^{(0)}}$ and hierarchical topics $\{\boldsymbol{\beta}^{(l)} \in \mathbb{R}^{D \times K^{(l)}}\}_{l=1}^L$ are represented as embedding vectors under the same semantic space, contributing to intuitively measuring and visualizing the distance between different topics (words), which has been proven to be effective in capturing the underlying semantic structure as shown in Fig. 5(a).

## 4 Policy Gradient-based Training Algorithm

### 4.1 ELBO of $dc$-ETM

As a VAE-like NTM, the developed $dc$-ETM can be trained like usual VAEs by directly maximizing the evidence lower bound (ELBO), specifically as

$$L(\boldsymbol{x}_n) = \mathbb{E}_{q(\boldsymbol{\theta}_n | \boldsymbol{x}_n)}[\ln p(\boldsymbol{x}_n | \boldsymbol{\theta}_n)] - \text{KL}(q(\boldsymbol{\theta}_n | \boldsymbol{x}_n) || p(\boldsymbol{\theta}_n)), \tag{7}$$

where the first term is the expected log-likelihood and the other term is the Kullback–Leibler (KL) divergence from the prior $p(\boldsymbol{\theta}_n)$ to the variational posterior $q(\boldsymbol{\theta}_n | \boldsymbol{x}_n)$.

Through introducing the augmented vectors $\{\boldsymbol{x}_n^{(l)}\}_{l=1}^L$, the log-likelihood of $\boldsymbol{x}_n$ in $dc$-ETM can be equivalently reformulated as

$$\ln p(\boldsymbol{x}_n | \boldsymbol{\theta}_n) = \mathbb{E}_{q(\{\boldsymbol{x}_n^{(l)}\}_{l=1}^L | -)} \left[\ln p(\boldsymbol{x}_n | \{\boldsymbol{x}_n^{(l)}\}_{l=1}^L) \prod_{l=1}^L p(\boldsymbol{x}_n^{(l)} | \boldsymbol{\theta}_n^{(l)})\right] \tag{8}$$

$$= \mathbb{E}_{q(\{\boldsymbol{x}_n^{(l)}\}_{l=1}^L | -)} \left[\ln p(\boldsymbol{x}_n | \{\boldsymbol{x}_n^{(l)}\}_{l=1}^L)\right] + \mathbb{E}_{q(\{\boldsymbol{x}_n^{(l)}\}_{l=1}^L | -)} \left[\sum_{l=1}^L \ln p(\boldsymbol{x}_n^{(l)} | \boldsymbol{\theta}_n^{(l)})\right],$$

where the function in the second expectation term can be treated as the summation of the set of log-likelihood of $\{\boldsymbol{x}_n^{(l)}\}_{l=1}^L$. Due to the hierarchical network structure, the KL divergence term can be factorized as

$$\text{KL}(q(\boldsymbol{\theta}_n | \boldsymbol{x}_n) || p(\boldsymbol{\theta}_n)) = \sum_{l=1}^L \mathbb{E}_{q(\boldsymbol{\theta}_n^{(l)} | -)} \left[\ln \frac{q(\boldsymbol{\theta}_n^{(l)} | -)}{p(\boldsymbol{\theta}_n^{(l)} | \boldsymbol{\Phi}^{(l+1)}, \boldsymbol{\theta}_n^{(l+1)})}\right], \tag{9}$$

where $q(\boldsymbol{\theta}_n^{(l)} | -)$ is constructed by a Weibull-based inference network described in Appendix A and $p(\boldsymbol{\theta}_n^{(l)} | \boldsymbol{\Phi}^{(l+1)}, \boldsymbol{\theta}_n^{(l+1)})$ satisfies a gamma prior in Eq. (1), and their KL divergence has an analytic expression, benefiting from adopting the Weibull reparameterization technique [20].

Combining the aforementioned derivations, the ELBO of $dc$-ETM can be equivalently rewritten as

$$L(\boldsymbol{x}_n) = \mathbb{E}_{q(\{\boldsymbol{x}_n^{(l)}\}_{l=1}^L | -)} \left[\ln p(\boldsymbol{x}_n | \{\boldsymbol{x}_n^{(l)}\}_{l=1}^L)\right] + \mathbb{E}_{q(\{\boldsymbol{x}_n^{(l)}, \boldsymbol{\theta}_n^{(l)}\}_{l=1}^L | -)} \left[\sum_{l=1}^L \ln p(\boldsymbol{x}_n^{(l)} | \boldsymbol{\theta}_n^{(l)})\right]$$

$$- \sum_{l=1}^L \mathbb{E}_{q(\boldsymbol{\theta}_n^{(l)} | -)} \left[\ln \frac{q(\boldsymbol{\theta}_n^{(l)} | -)}{p(\boldsymbol{\theta}_n^{(l)} | \boldsymbol{\Phi}^{(l+1)}, \boldsymbol{\theta}_n^{(l+1)})}\right], \tag{10}$$

which can be directly optimized with gradient-based methods to update both the encoder parameters $\boldsymbol{\Omega}$ and decoder parameters $\boldsymbol{\Psi}$ in $dc$-ETM. We emphasize that, after deriving the augmented vectors

$\{\boldsymbol{x}_n^{(l)}\}_{l=1}^L$ from $\boldsymbol{x}_n$ via data augmentation technique [8], the first expectation term in $L(\boldsymbol{x}_n)$ will be a constant and the ELBO can be directly optimized by maximizing the following loss function

$$
\begin{aligned}
\hat{L}(\{\boldsymbol{x}_n^{(l)}\}_{l=1}^L) &= \sum_{l=1}^L \mathbb{E}_{q(\boldsymbol{\theta}_n^{(l)}|-)}\left[\ln p(\boldsymbol{x}_n^{(l)}|\boldsymbol{\theta}_n^{(l)})\right] - \sum_{l=1}^L \mathbb{E}_{q(\boldsymbol{\theta}_n^{(l)}|-)}\left[\ln \frac{q(\boldsymbol{\theta}_n^{(l)}|-)}{p(\boldsymbol{\theta}_n^{(l)}|\boldsymbol{\Phi}^{(l+1)},\boldsymbol{\theta}_n^{(l+1)})}\right], \\
&= \sum_{l=1}^L \mathbb{E}_{q(\boldsymbol{\theta}_n^{(l)}|-)}\left[\ln \frac{p(\boldsymbol{x}_n^{(l)}|\boldsymbol{\theta}_n^{(l)})p(\boldsymbol{\theta}_n^{(l)}|\boldsymbol{\Phi}^{(l+1)},\boldsymbol{\theta}_n^{(l+1)})}{q(\boldsymbol{\theta}_n^{(l)}|-)}\right], \\
&= \sum_{l=1}^L \hat{L}^{(l)}(\boldsymbol{x}_n^{(l)};\alpha^{(l)},\hat{\boldsymbol{\Phi}}^{(l)},\boldsymbol{\theta}_n^{(l)},\{\boldsymbol{\Phi}^{(t)},\boldsymbol{\theta}_n^{(t)}\}_{t>l})
\end{aligned}
\tag{11}
$$

which can be roughly treated as the ELBO of a sequence $[\boldsymbol{x}_n^{(L)},...,\boldsymbol{x}_n^{(1)}]$ generated from a sequence of latent representations $[\boldsymbol{\theta}_n^{(L)},...,\boldsymbol{\theta}_n^{(1)}]$ [31, 32], and naturally meets the sequence-like generative process of $dc$-ETM as discussed in Sec. 3.2.

## 4.2 Optimization with Policy Gradient

Similar to RNN-based model, after augmenting $\{\boldsymbol{x}_n^{(l)}\}_{l=1}^L$ from $\boldsymbol{x}_n$, the loss function of $dc$-ETM defined in Eq. (11) is equal to the summation of $L$ sub-loss functions, where each sub-loss function $\hat{L}^{(l)}(\boldsymbol{x}_n^{(l)})$ can be equivalently regarded as a separate loss of a subsequence generation model that is only a part of the whole sequential generative model and expected to output $\boldsymbol{x}_n^{(l)}$ at the final time step $l$. Inspired by the great success achieved by RL methods [33, 34, 35, 36] in learning a stable long sequence (Markov decision process) with high quality, we consider the sequence-like generation procedure of a $L$-layer $dc$-ETM as a Markov decision process with $L$ time steps, and develop a novel training mechanism based on Policy Gradient [35] for $dc$-ETM, which injects the future rewards obtained from generating the suffix subsequence into each current sub-loss function $\hat{L}^{(l)}(\boldsymbol{x}_n^{(l)})$.

Specifically, we treat the whole $dc$-ETM as a stochastic policy network $\pi(a_n^{(l)}|s_n^{(l)})$ expected to generate a fixed-length action sequence $[a_n^{(L)},...,a_n^{(1)}]$ from the observation $\boldsymbol{x}_n$, defining the state $s_n^{(l)}$ as $\{\boldsymbol{x}_n,\{\boldsymbol{\Phi}^{(t)},\boldsymbol{\theta}_n^{(t)}\}_{t>l}\}$ and the action $a_n^{(l)}$ as $\alpha^{(l)}\hat{\boldsymbol{\Phi}}^{(l)}\boldsymbol{\theta}_n^{(l)}$. For each time step $l$, given the current state $s_n^{(l)}$, the policy network $\pi(a_n^{(l)}|s_n^{(l)})$ will first sample $\boldsymbol{\theta}_n^{(l)}$ from the inference network via

$$
\boldsymbol{\theta}_n^{(l)} \sim q(\boldsymbol{\theta}_n^{(l)}|\boldsymbol{x}_n,\{\boldsymbol{\Phi}^{(t)},\boldsymbol{\theta}_n^{(t)}\}_{t>l}),
\tag{12}
$$

and further obtain the corresponding action as

$$
a_n^{(l)} = \alpha^{(l)}\hat{\boldsymbol{\Phi}}^{(l)}\boldsymbol{\theta}_n^{(l)},
\tag{13}
$$

which can be regarded as directly drawing from $\pi(a_n^{(l)}|s_n^{(l)})$. The state transition is deterministic after an action has been chosen, indicating that the next state $s_n^{(l-1)} = \{\boldsymbol{x}_n,\{\boldsymbol{\Phi}^{(t)},\boldsymbol{\theta}_n^{(t)}\}_{t>l-1}\}$ if the current state $s_n^{(l)} = \{\boldsymbol{x}_n,\{\boldsymbol{\Phi}^{(t)},\boldsymbol{\theta}_n^{(t)}\}_{t>l}\}$ and the action $a_n^{(l)} = \alpha^{(l)}\hat{\boldsymbol{\Phi}}^{(l)}\boldsymbol{\theta}_n^{(l)}$.

Then we take the separate loss $\hat{L}^{(l)}(\boldsymbol{x}_n^{(l)})$ defined in Eq. (11) as the immediate reward at the the time step $l$, formulated as

$$
r(s_n^{(l)},a_n^{(l)}) = \mathbb{E}_{q(\boldsymbol{\theta}_n^{(l)}|-)}\left[\ln p(\boldsymbol{x}_n^{(l)}|\alpha^{(l)},\hat{\boldsymbol{\Phi}}^{(l)},\boldsymbol{\theta}_n^{(l)})\right] - \mathbb{E}_{q(\boldsymbol{\theta}_n^{(l)}|-)}\left[\ln \frac{q(\boldsymbol{\theta}_n^{(l)}|-)}{p(\boldsymbol{\theta}_n^{(l)}|\boldsymbol{\Phi}^{(l+1)},\boldsymbol{\theta}_n^{(l+1)})}\right],
\tag{14}
$$

and the action-value function can be formulated as

$$
Q^\pi(s_n^{(l)},a_n^{(l)}) = r(s_n^{(l)},a_n^{(l)}) + \mathbb{E}_\pi\left[\sum_{i=1}^{l-1}\gamma^i r(s_n^{(l-i)},a_n^{(l-i)})\right],
\tag{15}
$$

which indicates the expected accumulative reward starting from state $s_n^{(l)}$, taking action $a_n^{(l)}$, and then generating the suffix subsequence $[a_n^{(l-1)},...,a_n^{(1)}]$ with the policy network $\pi(a_n^{(l)}|s_n^{(l)})$ and the discount factor $0 < \gamma \le 1$.

Following [33], the objective function of training $dc$-ETM with policy gradient can be estimated (on one episode) as

$$J(\boldsymbol{x}_n; \boldsymbol{\Omega}, \boldsymbol{\Psi}) \simeq \sum_{l=1}^{L} \int_{a_n^{(l)}} \pi(a_n^{(l)}|s_n^{(l)}) Q^\pi(s_n^{(l)}, a_n^{(l)}) = \sum_{l=1}^{L} \mathbb{E}_{\pi(a_n^{(l)}|s_n^{(l)})} \left[ Q^\pi(s_n^{(l)}, a_n^{(l)}) \right], \tag{16}$$

where $\boldsymbol{\Omega}$ and $\boldsymbol{\Psi}$ indicate the encoder and decoder parameters in $dc$-ETM respectively. We note that the expectation $\mathbb{E}[\cdot]$ can be approximated by sampling methods based on the Weibull reparameterization, and the objective function can be directly optimized by advanced gradient descent algorithms, like Adam [37] and RMSprop [38]. We provide the details of PG-based training algorithm in Appendix C.

## 5 Experiments

To evaluate the effectiveness of the developed $dc$-ETM and the corresponding policy gradients (PG) based training algorithm, we make extensive experiments on both quantitative and qualitative aspects. Considering there are two $dc$-ETM variants as described in 3.1, we use the suffix $-\alpha$ and $-\beta$ to distinguish the variants defined in Eq. (2) and Eq. (3) respectively, and highlight whether the $dc$-ETM is trained with the PG based algorithm in Sec. 4.2. The implementation is available at https://github.com/yewen99/dc-ETM.

### 5.1 Datasets and Baselines

**Datasets:** Four widely used document benchmarks, specifically **R8** [39], 20Newsgroups (**20News**) [40], Reuters Corpus Volume I (**RCV1**) [41] and World Wide Web Knowledge Base (**WebKB**) [42] are included in the following experiments. We summarize the statistics of benchmarks in Appendix D and follow the procedure in [21] to preprocess these documents to obtain their BoW representations.

**Baselines:** We compare the developed $dc$-ETMs with a series of topic models, which can be roughly divided into two categories: 1) shallow topic models such as LDA [6], AVITM [12] and ETM [24], where LDA is a PTM and the others are NTMs; 2) deep topic models including PGBN [8], WHAI [20] and SawETM [21], where PGBN is a deep PTM and the others are deep NTMs. We emphasize that WHAI and SawETM are the most relevant strong baselines for comparison, both of which provide hierarchical Weibull-based latent document representations, and SawETM has achieved state-of-the-art performance on unsupervised document modeling and clustering tasks.

**Experimental Settings:** To make a fair comparison, we set the same network structure for all deep topic models as $[256, 128, 64, 32, 16]$ from shallow to deep. For PTMs, we use the default hyperparameter settings in their published papers and accelerate the Gibbs sampling with GPU. For NTMs, we set the size of their hidden layers as 256, the embedding size as 100 for them incorporating word embeddings, like ETM, SawETM and $dc$-ETMs, and the mini-batch size as 200. For optimization, we adopt the same Adam optimizer [43] with a learning rate of 1e-2. All experiments are performed with an Nvidia RTX 3090 GPU and implemented with PyTorch [44].

### 5.2 Quantitative Comparisons

To investigate whether the proposed skip-connection structures in both $dc$-ETM variants can alleviate "*posterior collapse*", especially at higher layers, we compare them with other popular deep topic models in the first part. Then, to investigate whether the mitigation of "*posterior collapse*" can improve the quality of latent document representations at higher layers, we conduct more quantitative comparisons in the rest parts. We report the error bars in Appendix E.

**Document Modeling:** In Fig. 2, for each deep topic model, we plot the curve of point log-likelihood $\ln p(\boldsymbol{x}_n|-)$ as a function of iterative epochs conditioned on the $t$-th-layer reconstruction $\hat{\boldsymbol{\Phi}}^{(t)} \boldsymbol{\theta}_n^{(t)}$, which can be used to measure the degree of "*posterior collapse*" by the relevance between the data sample $\boldsymbol{x}_n$ and its latent representation $\boldsymbol{\theta}_n^{(t)}$. From the results, we can see that although SawETM and WHAI achieve a comparable performance with $dc$-ETMs on the first hidden layer in Fig. 2(a), their reconstruction quality decreases dramatically with the network going deeper in Fig. 2(b) and 2(c), potentially reflecting that little data information can be propagated to higher layers of these traditional deep topic models. Benefiting from introducing skip connections into the generative process, $dc$-ETMs can significantly alleviate "*posterior collapse*" at higher hidden layers.

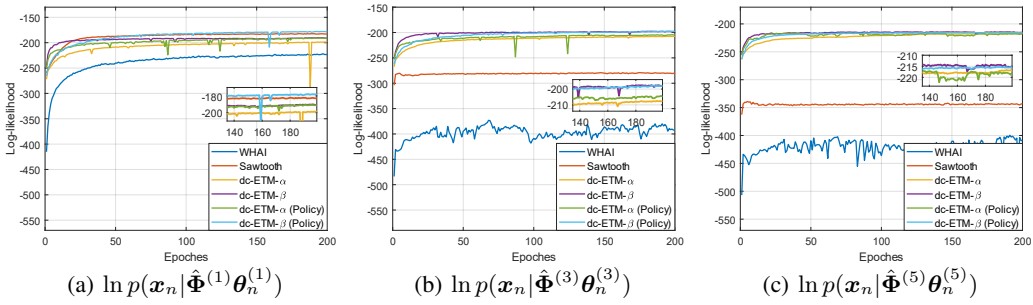

(a) $\ln p(\boldsymbol{x}_n|\hat{\boldsymbol{\Phi}}^{(1)}\boldsymbol{\theta}_n^{(1)})$      (b) $\ln p(\boldsymbol{x}_n|\hat{\boldsymbol{\Phi}}^{(3)}\boldsymbol{\theta}_n^{(3)})$      (c) $\ln p(\boldsymbol{x}_n|\hat{\boldsymbol{\Phi}}^{(5)}\boldsymbol{\theta}_n^{(5)})$

Figure 2: Point log-likelihood $\ln p(\boldsymbol{x}_n|\hat{\boldsymbol{\Phi}}^{(t)}\boldsymbol{\theta}_n^{(t)})$ of different deep topic models on 20News dataset as a function of iterative epoches, where $\hat{\boldsymbol{\Phi}}^{(t)}\boldsymbol{\theta}_n^{(t)}$ can be treated as the projection of $\boldsymbol{\theta}_n^{(t)}$ from the latent space to the observation space, as discussed in Sec. 3.2.

**Perplexity & Topic Diversity:** To make a more comprehensive quantitative comparison, we use the average of heldout-word perplexities (the lower is the better) and topic diversities (the higher is the better) across all hidden layers to measure the document modeling performance and topic quality of these deep topic models with $\{\boldsymbol{\theta}_n^{(t)}\}_{t=1}^T$ and $\{\boldsymbol{\Phi}_n^{(t)}\}_{t=1}^T$, respectively. The experimental settings are consistent with those in [21], and the experimental results have been exhibited in Table 3. Benefiting from hierarchical network structures, the modeling capability of deep topic models generally outperform those shallow ones. Thanks to enhancing the connections between the observation and multiple hidden layers with the deep-coupling generative process, the developed $dc$-ETMs achieve lower perplexity scores and provide higher-quality topics than traditional topic models. Then, the PG-based training algorithm brings further performance improvement to our $dc$-ETMs.

Table 1: Comparisons of the average of perplexities and topic diversities across all hidden layers on various benchmarks.

| Model | Perplexity | | | Topic Diversity | | |
|---|---|---|---|---|---|---|
| | R8 | 20News | RCV1 | R8 | 20News | RCV1 |
| LDA [6] | 996 | 1091 | 1242 | 0.288 | 0.356 | 0.423 |
| AVITM [12] | 561 | 1030 | 1121 | 0.330 | 0.408 | 0.483 |
| ETM [24] | 985 | 989 | 1480 | 0.352 | 0.410 | 0.524 |
| PGBN [8] | 657 | 743 | 1086 | 0.221 | 0.186 | 0.355 |
| WHAI [20] | 773 | 870 | 1192 | 0.183 | 0.158 | 0.294 |
| SawETM [21] | 530 | 732 | 920 | 0.207 | 0.175 | 0.331 |
| $dc$-ETM-$\alpha$ | 521 | 730 | 912 | 0.212 | 0.281 | 0.435 |
| $dc$-ETM-$\beta$ | 427 | 710 | 873 | 0.346 | 0.429 | 0.566 |
| $dc$-ETM-$\alpha$ (Policy) | 463 | 707 | 896 | 0.279 | 0.385 | 0.519 |
| $dc$-ETM-$\beta$ (Policy) | **420** | **647** | **841** | **0.379** | **0.456** | **0.584** |

**Document Clustering:** To evaluate the quality of the extracted latent document representations on downstream tasks, we consider document clustering, where we use the topic models after training to extract the latent representations of the testing documents and then use k-means to predict the clustering labels. Using the Purity and Normalized Mutual Information (NMI) as metrics (the higher the better), the results shown in Table 4 demonstrate that concatenating hierarchical latent document representations extracted by traditional deep topic models cannot improve and even hurt the clustering performance, potentially indicating that the latent representations at higher layers are meaningless. However, distinct from traditional deep

Table 2: Document clustering comparison on the 1st hidden layer or the concatenation of all hidden layers of different topic models.

| Model | Layer | WebKB | | 20News | | R8 | |
|---|---|---|---|---|---|---|---|
| | | Purity | NMI | Purity | NMI | Purity | NMI |
| LDA | 1 | 53.40 | 11.23 | 41.79 | 45.15 | 65.74 | 40.47 |
| AVITM | 1 | 54.18 | 17.77 | 42.33 | 46.33 | 70.96 | 41.20 |
| ETM | 1 | 51.43 | 12.52 | 42.61 | 48.40 | 72.20 | 41.28 |
| PGBN | 1 | 55.37 | 16.27 | 43.30 | 46.51 | 74.52 | 41.24 |
| | All | 53.58 | 15.39 | 41.17 | 44.20 | 72.93 | 31.35 |
| WHAI | 1 | 59.89 | 25.95 | 42.25 | 46.98 | 74.70 | 43.98 |
| | All | 57.46 | 24.49 | 32.00 | 37.51 | 70.80 | 41.25 |
| SawETM | 1 | 57.89 | 21.91 | 43.33 | 50.77 | 75.25 | 42.97 |
| | All | 51.75 | 20.60 | 38.69 | 39.33 | 75.89 | 39.55 |
| $dc$-ETM-$\alpha$ | 1 | 61.14 | 26.29 | 32.81 | 43.64 | 75.60 | 39.83 |
| | All | 63.18 | 28.35 | 41.83 | 44.52 | 76.31 | 43.73 |
| $dc$-ETM-$\beta$ | 1 | 54.71 | 21.43 | 39.80 | 44.30 | 74.30 | 38.63 |
| | All | 67.29 | 33.60 | 45.00 | 46.20 | 76.25 | 45.64 |
| $dc$-ETM-$\alpha$ (Policy) | 1 | 49.71 | 14.86 | 37.88 | 43.56 | 71.65 | 32.73 |
| | All | 64.32 | 33.65 | 42.21 | 45.59 | 77.46 | 44.60 |
| $dc$-ETM-$\beta$ (Policy) | 1 | 57.32 | 26.05 | 40.11 | 44.12 | 71.30 | 38.34 |
| | All | **69.32** | **38.53** | **48.60** | **55.79** | **78.29** | **48.62** |

| 5_0: game team games hockey baseball play year players season fans |
| 5_1: host nntp posting lines subject organization distribution mit world access |
| 5_2: com article writes apr lines subject organization netcom reply mark |
| 5_3: max israel turkish jews armenian armenians war Israeli jewish armenia |
| 5_4: president national states health american press united cliton year april |
| 5_5: gun people government right law rights guns state fbi weapons |
| 5_6: god jesus bible Christian people believe church truth say know |

(a) $\hat{\boldsymbol{\Phi}}^{(5)}$ learned by $dc$-ETM

| 5_0: lines subject organization com article just don writes university like |
| 5_1: lines subject organization com article just don university writes like |
| 5_2: lines subject organization com article just don writes university like |
| 5_3: lines subject organization com article just don writes university like |
| 5_4: lines subject organization com article just don university writes like |
| 5_5: lines subject organization com article just don university writes like |
| 5_6: lines subject organization com article just don writes university like |

(b) $\hat{\boldsymbol{\Phi}}^{(5)}$ learned by SawETM

Figure 3: The 5th-layer topics learned by $dc$-ETM and SawETM with the same network structure on 20News, where each topic is interpreted by its top-10 words. More comparisons refer to Appendix F.

topic models, the concatenation operation on the latent representations of $dc$-ETMs can significantly improve the performance, which can be attributed to enforcing strong links between the multi-layer representations and the observation with the skip connections in the generation.

### 5.3 Qualitative Analysis

As discussed in Sec. 3.2, the developed $dc$-ETM inherits both the characteristics of hierarchical topic structure and semantic topic embeddings. Then we compare the hierarchical topics of a 5-layer $dc$-ETM trained on 20News with those learned by SawETM for qualitative analysis.

**Topic Visualization:** With the visualization techniques [8], we exhibit the 5th-layer topics learned by $dc$-ETM and SawETM on 20News in Fig. 3 and Fig. 7, where each topic is interpreted by its top-10 words by sorting the word probabilities by descending order. Obviously, the topics learned by SawETM are quite similar, explaining the reason why concatenating its hierarchical latent document representations cannot improve and even hurt the performance on downstream tasks. On the contrary, the developed $dc$-ETM can learn meaningful and diverse topics at higher layers, indicating that more data information is passed to higher layers to alleviate "*posterior collapse*". We also exhibit a 5-layer topic tree learned by $dc$-ETM in Fig. 7 to illustrate the topic hierarchy of $dc$-ETM in Appendix N.

**Topic Embedding Visualization:** After extracting hierarchical topic trees by $dc$-ETM, we visualize some of these trees originated from different topic nodes at layer 5 by projecting their semantic embeddings with t-SNE [45]. As shown in Fig. 5(a), we can find that the topics in the same topic tree tend to be closer than others from different trees in the semantic embedding space and similar phenomenon occurs in the words for describing the same root topic, which indicates the hierarchy learn by $dc$-ETM is of high quality. Note that we also visualize the topics consisting of similar top words in Fig. 5(b) and 5(c), learned by $dc$-ETM and SawETM respectively, which demonstrates that the developed $dc$-ETM can provide more meaningful and discriminative topic and word embeddings.

## 6   Conclusion

To provide higher-quality hierarchical latent representations for deep topic modeling, in this paper, with the deep-coupling generative process, we develop a novel $dc$-ETM, which is constructed by introducing skip connections into the generative process of GBN and also incorporates both topic

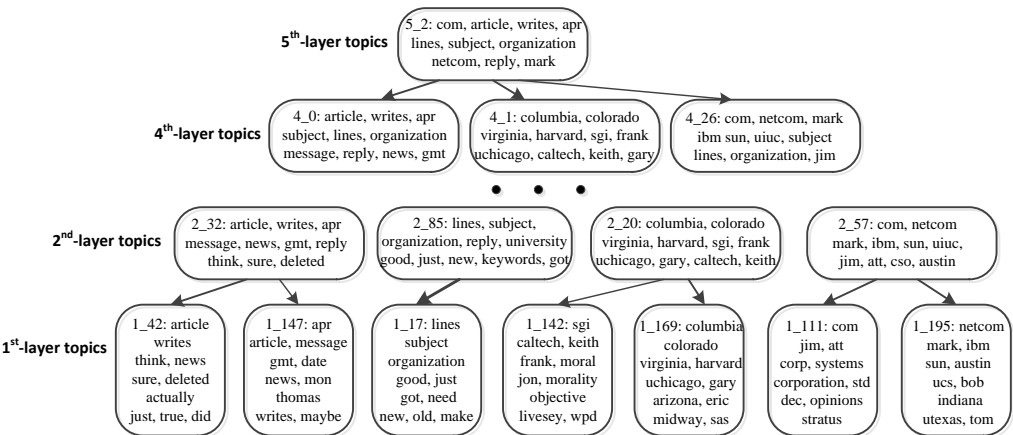

Figure 4: A hierarchical topic tree example learned by a 5-layer *dc*-ETM on 20News dataset.

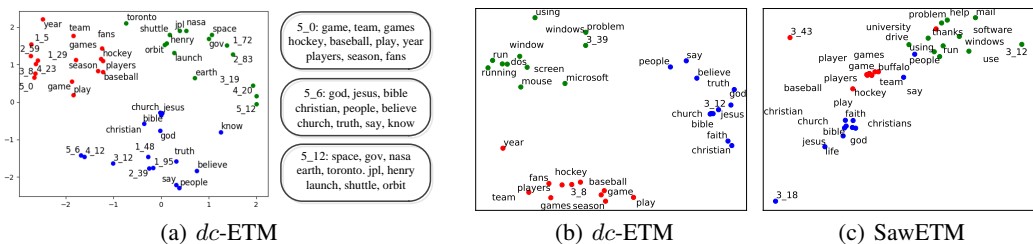

(a) *dc*-ETM      (b) *dc*-ETM      (c) SawETM

Figure 5: t-SNE visualization of a) multiple 5-layer hierarchical topic trees learned by *dc*-ETM, whose leaf nodes are distinguished by different colors; b) and c) various semantic topics equipped with their own top-10 representative word embeddings learned by *dc*-ETM and SawETM on 20News.

embedding and Weibull reparameterization techniques. Utilizing the property of sequence-like generation process, we design a PG-based training algorithm for *dc*-ETM to further alleviate the information reduction at higher layers. We note that the main idea of designing *dc*-ETM equipped with the PG-based training algorithm can potentially be extended to other deep topic models.

## Acknowledgments

This research is supported in part by the National Research Foundation, Singapore under its Industry Alignment Fund – Pre-positioning (IAF-PP) Funding Initiative. Any opinions, findings and conclusions or recommendations expressed in this material are those of the author(s) and do not reflect the views of National Research Foundation, Singapore. Additionally, this work is supported in part by the National Natural Science Foundation of China under Grant U21B2006; in part by Shaanxi Youth Innovation Team Project; in part by the 111 Project under Grant B18039; in part by the Fundamental Research Funds for the Central Universities QTZX22160.

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
