# OpenReview forum: "Alleviating "Posterior Collapse'' in Deep Topic Models via Policy Gradient"
_NeurIPS.cc/2022/Conference — NeurIPS 2022 Accept_

### Official Review · Reviewer_JvJD · 2022-07-09

**Rating:** 6
**Confidence:** 3
**Soundness:** 2 fair
**Presentation:** 3 good
**Contribution:** 4 excellent

**Summary:**

A new hierarchical neural topic model is proposed that addresses the posterior collapse that occurs at higher levels with previous hierarchical neural topic models. The resulting topic model achieves much higher quality topics at higher levels. The method can be extended to different types of neural topic models.

**Questions:**

- How does the method perform in comparison to traditional deep topic models like nHDP?
- What does the comparison look like with respect to topic coherence?
- Why are the Table 2 results only performed on the first hidden layer representations?
- Is there some way to evaluate the hierarchy level-wise to be able to make quantitative claims about the topic quality at higher levels?

**Limitations:**

limitations are not addressed

**Strengths And Weaknesses:**

# originality
The work has a high degree of novelty. The augmented representation of the documents which allows to reduce the posterior collapse is very interesting.

# quality
The claims are in general well supported. Nevertheless, the experimental section could be more convincing. It is uncommon nowadays to compare topic models with regard to perplexity though. It has been shown that perplexity does not correlate well with human judgment. Usually, topic quality, a product of diversity and coherence is used instead. For hierarchical topic models, it is even possible to evaluate the different layers separately since the coherence and diversity could be much different depending on the layer. There is a recent attempt in that direction which could be helpful in comparing different hierarchies [1]. The claim that the resulting topics have higher quality is hard to support with only perplexity and diversity measures.

# clarity
The paper is well written and structured.

# significance
The method is likely to be used and further developed. It is extendable to different deep topic models and thus has general applicability.


minor comments:
- Figure 1: a) says Deep PGM but the description says Deep PTM
- line 129 alleviate -> alleviates


[1] Nagda et al. "Hierarchical Topic Evaluation: Statistical vs. Neural Models" Bayesian Deep Learning Workshop at NeurIPS 2021.

---

> ### Author Response · Authors · 2022-08-02
> **Response to Reviewer JvJD**
>
> Thanks for your effort in reviewing this paper!
>
> **For quality: topic quality, a product of diversity and coherence is used instead.**
>
> Thanks for your valuable suggestion! We agree with your point that it will be better to measure the layer-wise coherence and diversity for hierarchical topic models and have included the corresponding experimental results in Appendix K.
>
> **For the "minor comments"**
>
> Thanks for your notification. We have fixed these typos.
>
> **For Question 1**
>
> Thanks. It is a great suggestion to include more traditional PTMs and we are very glad to compare our methods with them. We will try to update the related experimental results in the next few days if we can.
>
> **For Question 2**
>
> Thanks. Please refer to Appendix K for detailed experimental results.
> From the results, we can see that our method could not only achieve comparable performance on topic coherence but also obtain higher gains on the aspect of topic quality, which is calculated by the product of topic diversity and topic coherence.
>
> **For Question 3**
>
> We note that we also included the results measured on the concatenation of all hidden layers, termed as "ALL''.
> The reason why we choose the first hidden layer for comparison is that traditional deep topic models usually use the first hidden layer, which could be the most informative document representation, to perform downstream tasks.
> And the concatenation of hierarchical latent document representations will even hurt their model performance, which is mainly caused by "posterior collapse''.
> On the contrary, from the results in Table.2, we can see that our method can significantly improve the performance of the concatenation of hierarchical latent document representations, term as 'ALL'.
>
> We note that we have also provided the comparisons on other hidden layers as shown in Appendix L.
>
> **For Question 4**
>
> Thanks. We think that it is a good suggestion to evaluate the hierarchy level-wise. However, it seems that there is no existing metric to measure it as far as we know.
> If you have further comments, we are glad to test our methods with these metrics.

---

### Official Review · Reviewer_1MCV · 2022-07-10

**Rating:** 5
**Confidence:** 4
**Soundness:** 3 good
**Presentation:** 2 fair
**Contribution:** 2 fair

**Summary:**

This paper presents a hierarchical neural topic model (NTM) along with
a policy gradient approach for learning the parameters. The core of
the NTM extends existing work (Embedding Topic Model: ETM). Learning
occurs within a variational autoencoder framework. To do this
formulation, the hiearchical parameters are treated as the actions
that must be produced, with the corresponding expectations of the ELBO
as the reward function. The paper performs quantitative and
qualitative comparisons on four datasets, comparing against six
baselines. The paper reports gains in perplexity, topic diversity,
topic purity, and NMI.

**Questions:**

Q1. Please provide a brief explanation of W1 that could be added to a
revised paper.

Q2. Please provide a brief explanation of W2 that could be added to a
revised paper.

Q3. Please address W3.

Q4. Please provide a brief explanation of W4 that could be added to a
revised paper.

Q5. Please provide a brief explanation of W5 that could be added to a
revised paper.

Q6. How many samples for the VAE objective (Eq. 11) or the policy
gradient objective (Eq. 16)?


**Limitations:**

No, see W1-W4.

Post discussion: sufficiently addressed

**Strengths And Weaknesses:**

UPDATE post discussion: I thank the authors for their clarifications and promises to revise accordingly. I have updated my main S/W review. I am also updating my score.

=================================

From the very beginning of this paper, there is a focus on "posterior
collapse." However, the paper does not adequately or properly define
posterior collapse. This is a major weakness of the paper, and would
require significant revisions throughout. Notably:

W1. How is posterior collapse defined? [Post discussion: thank you for addressing this. I strongly recommend putting the revised definition in the main paper.]

W2. How is posterior collapse being measured? [Post discussion: thank you for addressing this. I strongly recommend clearly tying this to the experimental setup in the main paper.]

W3. What experimental results demonstrate that posterior collapse, as
measured by point (2), has actually been mitigated? [Post discussion: Please add the revised table in the appendix to the main paper.]

Beyond that overarching weakness, this paper does not adequately
address other considerations, including broader take-aways:

W4. This paper presents dc-ETM, and two learning algorithms for
it. What aspects of this work prevent posterior collapse? Is it the
definition of dc-ETM, the policy gradient approach, or both? [Post discussion: thank you for addressing this]

W5. Fundamentally, why does policy gradient result in better results?
The two objectives (Eq. 11 and Eq. 16) are very similar, sharing the
same subloss computations. From what I can tell, a core difference is
the quadratic nature of the policy gradient objective. Is this what is
responsible for the improved results, or is there something else at
play?

One strength of this work is that it examines different ways of
decoding, notably through Eqns 2 and 3.

W6. Unfortunately, the presentation of the model(s) needs to be
improved. While some of this could be a spacing issue, e.g., around
equation 1), the corresponding prose is quite dense and more difficult
to get through than it should be. Further, the core factorization and
probabilistic identities that this paper relies (e.g., the
deconstruction in Eq. 4) on are not specified clearly. I recommend
significantly rewriting and streamlining the presentation of the model
and underlying justifications. [Post discussion: thank you for addressing this]

W7. Finally, I found the qualitative results to be unclear and
difficult to interpret. Figure 5 does not have enough explanation,
either in the figure itself, the caption, or the corresponding prose.
[Post discussion: thank you for addressing this]

---

> ### Author Response · Authors · 2022-08-02
> **Response to Reviewer 1MCV (1/2)**
>
> Thanks for your effort in reviewing this paper and we apologize for your misunderstanding of the purpose of this work due to our lacking of explanation for "posterior collapse''.
> We note that we have included clear and enough preliminary knowledge about the ``posterior collapse'' in the revision, and we hope you can try to capture the main idea of this work, whose novelty has been acknowledged by the other two reviewers. And we are glad to have a discussion with you in the following days.
>
> Here, please allow us to explain the main idea and novelty of our work.
>
> At first, we need to emphasize that this work is developed to mitigate the phenomenon of "posterior collapse'' in deep topic models, which has been widely disclosed in the topic modeling literature but few efforts have been made to address this challenge.
> The main difficulty is the need for carefully designing the probabilistic generative process to build an effective connection between the observation and the corresponding latent representations, on the premise of preserving the interpretable hierarchical topic modeling structure, rather than casually introducing skip-connections.
> Utilizing the natural hierarchy in deep topic models, we provide a general solution equipped with a novel perspective to incorporate RL-based training algorithm, which has been recognized by **Reviewers kqfN** and  **JvJD** for bringing quite a few contributing ideas to this field.
>
>
> The main idea of this work can be applied to extend NTMs with similar structures. Most of the existing NTMs adopt VAE-liked structures, but none of them attempt to solve "posterior collapse'' phenomenon in essence, resulting in that the latent representations at higher layers exhibit similar or meaningless patterns (as shown in the Appendix).
> The significance of developing deep-coupling structure with RL-based training algorithm, which
> has effectively improved the quality of the latent representations of a deep topic model at higher layers, going beyond a single specific model.
>
> **For W1**
>
> For a VAE-based model consisting of a decoder $p_\theta(x|z)$ and an encoder $q_\phi(z|x)$, the definition of posterior collapse is that the posterior of latent variables, denoted as $q_\phi(z|x)$, collapses to it prior $p_\theta(z)$, which is a non-informative distribution and independent of the data $x$.
> It can be mathematically denoted as that the KL divergence between $q_\phi(z|x)$ and $p_\theta(z)$ is close to zero, represented as $D_\{KL\} [q_\phi(z|x)||p(z)] \approx 0$.
>
> We have explained "posterior collapse'' in Line 35-44 of the revision, and also added a theoretical explanation in **Appendix J** to explain why posterior collapse would happen, which could help the readers understand why we want to introduce skip-connections into deep topic models.
>
> **For W2**
>
> Posterior collapse can be directly measured by the relevance between the data distribution $p(x)$ and the posterior of latent variables $q_\phi(z|x)$.
> Thus, for a VAE-baed model, a straightforward metric to measure the degree of posterior collapse could be the likelihood score, denoted as $p_\theta(x|z)$, where $z$ is drawed from its posterior $q_\phi(z|x)$.
> As the results shown in Fig. 2, the point log-likelihood scores of deep NTMs (WHAI and SawETM) decrease rapidly as the network goes deeper, potentially reflecting that these traditional deep topic models suffer from serious "posterior collapse'' and little data information can be propagated to their higher layers.
>
> As the response to W1, another promising metric to measure "posterior collapse'' could be the KL-divergence between $q_\phi(z|x)$ and $p_\theta(z)$, where a smaller KL-divergence score indicates that the posterior $q_\phi(z|x)$ contains less data information and has a larger tendency to collapse to its non-informative prior.
> In such consideration, we provide an additional experiment of the KL-divergence comparison for SawETM and dc-ETM in **Table 5 of Appendix J**.
>
> Posterior collapse can also be potentially evaluated by those metrics of applying the latent variables sampled from $q_\phi(z|x)$ for downstream tasks.
> To make a comprehensive comparison of the degree of posterior collapse between SawETM and our method, we have also included the comparison of topic coherence and clustering performance at different hidden layers learned by these models as shown in **Appendix K and Appendix L**.

---

> > ### Author Response · Authors · 2022-08-02
> > **Response to Reviewer 1MCV (2/2)**
> >
> > **For W3**
> >
> > As the response to W2, the experimental results in Fig.2 could be the most straightforward way to illustrate that our method can effectively alleviate "posterior collapse''.
> >
> > From the results, we can see that although SawETM and WHAI achieve comparable performance with $dc$-ETMs on the first hidden layer in Fig. 2(a), their reconstruction quality decreases dramatically with the network going deeper in Fig. 2(b) and 2(c), potentially reflecting that little data information can be propagated to higher layers of these traditional deep topic models.
> > Benefiting from introducing skip connections into the generative process, $dc$-ETMs can significantly alleviate the posterior collapse at higher layers.
> >
> > **For W4**
> >
> > Thanks. Both of these two aspects contribute to providing more expressive latent representations by alleviating posterior collapse. Intuitively, the introduced physical skip-connection structures in dc-ETM can straightforwardly build the connections between the observation $x_n$ and its higher-level latent representations {$\theta_n^\{(l)\}$},${l>1}$, resulting in that the data information can be directly passed to these hidden layers to alleviate "posterior collapse''.
> > As the response to W5, applying PG-based method for model training can also alleviate posterior collapse by providing more informative latent representations in a virtual way.
> >
> >
> >
> > **For W5**
> >
> > The motivation of using Policy Gradients is that, directly training a topic model with EBLO can not update the model parameters until the whole sequence is generated, which is more likely to generate an in-balanced sequence of bad performance [31].
> > Considering RL is a well-known method for learning a long sequence (Markov Decision Process) of good quality,
> > we choose to apply the Policy Gradients, which is one of the most stable algorithms in RL, for training the developed dc-ETM.
> >
> > The potential reason behind the success of applying Policy Gradients for model training is that, it can decouple the target of the whole sequence generation into a series of generation of subsequence, and then the training objective at each time step will not only focus on the generation quality at the current time step, i.e., $r_t$ in Eq. (14), but also consider the future rewards collected from the following time steps $E_\pi\left[\sum_\{i=1\}^\{l-1\}\gamma^i r(s_n^\{(l-i)\}, a_n^\{(l-i)\})\right]$ in Eq. (15).
> >
> > Thus, with PG-based training algorithms, we can empirically force the latent variables $\theta_n^{(l)}$ at each hidden layer to collect more data information from those at lower layers, specifically $\theta_n^\{(l-1)\}, ..., \theta_n^\{(1)\}$, and further improve the quality of these latent document representations learned by dc-ETM, leading to better model performance on downstream tasks.
> >
> >
> > [31] Seqgan: Sequence generative adversarial nets with policy gradient.
> >
> > **For "One strength of this work is that it examines different ways of decoding, notably through Eqns 2 and 3."**
> >
> > Thanks, we have also added an illustration of these two ways of decoding in Appendix I for a better understanding of the two different projections ($\alpha$ and $\beta$).
> >
> > **For W6**
> >
> > Thanks for your careful reading! We have revised several places to make our paper easier to be followed and promise that we will follow your suggestions to revive our paper.
> >
> > **For W7**
> >
> > Thanks, actually, we have interpreted the results of Fig. 5 in Section 5.3 named "Topic embedding visualization'' in the first manuscript.
> > Moreover, we have included more explanations for qualitative results in the revision.
> >
> > **For Q6**
> >
> > Only one sample is enough, which is efficient.
> > A similar conclusion can be found in [1],
> > Kingma et al have also demonstrated that one sample per iteration is still enough when the batch size is large.
> >
> > [1] Kingma et al. Auto-Encoding Variational Bayes.

---

> > > ### Comment · Reviewer_1MCV · 2022-08-09
> > > **Reponse**
> > >
> > > Thank you for sufficiently addressing W3, W4, W6, and W7.
> > >
> > > For W5, it appears that this *structured* look-ahead is a core reason why the proposed method works. Is that correct?

---

> > > > ### Author Response · Authors · 2022-08-09
> > > > **For W5**
> > > >
> > > > Thanks for your carefully reviewing our paper again.
> > > >
> > > > It is correct and we think you have captured the main thought of applying PG-based method for training our models.  As explained in our response to W5, similar to the recent popular idea of applying RL for sequence generation, the core reason why PG-based training algothrim will work is that the training objective at each time step $t$ will consider the future rewards (training objectives) collected from the following time steps $t+1, ..., L$, rather than only focusing on the objective at the current time step, which can improve the quality of the whole generation sequence and provide more expressive latent representations for downstream tasks.
> > > >
> > > > We note that our work is the first one to treat the typical top-down generative process of hierarchical topic model ``$  \theta_n^\{(L)\}\rightarrow \theta_n^\{(L-1)\}\rightarrow ... \rightarrow \theta_n^\{(1)\} \rightarrow x_n$'' as a sequence generation task, which can be formulated as a Markov decision process and then RL-based methods can be naturally applied for model training as discussed in Section 4.2.
> > > >
> > > > Then, applying RL-based method on topic models is also one of the main contributions of our work and has been admited by the other two reviewers that our work is off sufficient interests to the topic modeling community.
> > > >
> > > > Thanks for your spending time to have a discussion with us
> > > >
> > > > Best wishes
> > > >
> > > > Authors

---

> ### Author Response · Authors · 2022-08-07
> **Response to Reviewer 1MCV (Further Discussion)**
>
> Dear Reviewer 1MCV,
>
> Following your constructive suggestions，we have revised our paper and provided an additional section to introduce posterior collapse in **Appendix J** for better understanding our paper's main idea，including the **definition** of posterior collapse，how to **measure** posterior collapse and **why** posterior collapse would happen in VAE-based models.
>
> We tried our best to address your concerns. Are there unclear explanations here? We could further clarify them.
> We are willing to have a discussion with you in the following days!
>
>
> Best regards,
>
> Authors

---

> > ### Comment · Reviewer_1MCV · 2022-08-09
> > **Importance of posterior collapse defn & KL**
> >
> > Thank you for providing this information. When it comes to *posterior collapse*, I find these KL experiments a much needed and critical inclusion. I strongly believe they need to be part of the main paper. Why? The approach used in this paper involves learning both an encoder and decoder: the point LL results in Figure 2 demonstrate that a better decoder has been learned, rather than directly addressing the posterior approximation itself.
> >
> > Further questions/recommendations for clarification:
> > 1. can you clarify how you are computing KL?
> > 2. what's the variance on the KL terms?

---

> > > ### Author Response · Authors · 2022-08-09
> > > **Thanks for your feedback**
> > >
> > > Thanks for your feedback.
> > >
> > > **Q1: can you clarify how you are computing KL?**
> > >
> > > As Eq. (25) shown in Appendix, the KL divergence between a Weibull distribution and a Gamma distribution is analytic, which can be directly calculated after obtaining $k$ and $\lambda$ in Weibull distribution with the inference network shown in Eq. (22), and $\alpha$ and $\beta$ in a Gamma prior distribution defined in Eq. (1).
> > >
> > > We note that we have claimed these details in Line 173 - 175 in the manuscript, **which was also included in our first submission.**
> > >
> > > **Q2: what's the variance on the KL terms?**
> > >
> > > For each given data sample $x$, the KL divergence is a scalar without variance, and thus we can directly calculate the KL term at each hidden layer $l$ with its defination $E_{q(\theta_n^\{(l)\} | -)}[\ln (q(\theta_n^\{(l)\}| -) / p(\theta_n^\{(l)\} | \phi^\{(l+1)\}, \theta_n^\{(l+1)\}))]$, which is in the same way with the procedure of calculating the KL term in ELBO.
> > >
> > > If you are interested in finding out the variance on layer-wise KL terms of different data samples, we are glad to include them in our revision, but not this turn due to the limited time. In our experience, the variance will be  relatively small when compared to these KL terms.
> > >
> > > **Suggetions ``I strongly believe they need to be part of the main paper''**
> > >
> > > Thanks, we agree with the point that KL divergence to measure the relavance between $q_\phi(z|x)$ and $p_\theta(z)$ could be a more straighforward way to measure posterior collpase and will follow your suggestions to include these experimental results in our revision.
> > > However, we still need to point out that the point LL results in Figure 2 can be used to measure the relavance between $q_\phi(z|x)$ and $p(x)$, which reflects the amout of data information contained in these latent variables $z$ and can also measure the degree of ``posterior collapse''.
> > >
> > > Thanks, please let us know if you have further questions.
> > >
> > > Best wishes
> > >
> > > Authors

---

> ### Author Response · Authors · 2022-08-09
> **Further Discussion**
>
> Dear Reviewer 1MCV:
>
> Thanks again for your effort in reviewing our paper and give us a great chance to improve the paper quality.
>
> Considering that the discussion period is coming to an end, we would like to know if you have any other questions about our paper, and we are still glad to have a discussion with you in the limited time.
> If you do not have time to reply us, perhaps you can discuss with the other two reviewers after the discussion period, who have captured the main idea of our work and judge it as a high-quality draft.
>
> Sorry for disturbing you again and again, we only want to let you know your decision is quite important to us.
>
> Sincerely
>
> Authors

---

### Official Review · Reviewer_kqfN · 2022-07-11

**Rating:** 6
**Confidence:** 4
**Soundness:** 3 good
**Presentation:** 2 fair
**Contribution:** 3 good

**Summary:**

The authors develop a deep-coupling generative process for deep topic modeling, where the key idea is to incorporate skip connections into the generation to alleviate the issue that higher layer representations exhibit similar patterns. The authors further formulate the sequence-like generation procedure as a Markov decision process, and develop a Policy Gradient based (PG-based) learning method. The authors show the benefit of PG-based training over vanilla training for topic modeling.

In comparison with previous methods, the experimental results show the consistent and significant benefits on three datasets. The qualitative evaluation (i.e., looking into generated topics and topic embeddings) also support the authors claim (overcoming ``posterior collapse”).

**Questions:**

1. The clear performance gap on dc-etm-alpha (Equation 2) and dc-etm-beta (Equation 3) does seem to be big. The authors can shed some light on it.

2. Regarding PG-based vs non-PG-based -- Table one/two do support that PG-based training is helpful. However, more analysis/discussions could be helpful:

    2.1. What is the motivation to use PG-based training, besides observing the success in [25], [26] and [27].

    2.2 According to Figure 2, PG-training does not seem to benefit point log-likelihood, but in Table 1 and 2, PG-based models exhibit consistent benefits, can the authors discuss more on this "discrepancy".

    2.3 Can the authors show qualitative comparison (e.g., t-sne plot of topic embeddings) between dc-ETM-alpha, dc-ETM-beta and dc-ETM-beta (Policy).

3. Regarding hierarchical representation:

    3.1. The table two does show that the higher-layer representations of PGBN/WHAI/SawETM do not help with documentation clustering; Interestingly, the 1st layer hidden layer of PGBN/WHAI/SawETM does seem to outperform dc-ETM.

    3.2. Can the authors compare the 4th/3rd/2nd/1st layer visualization for SawETM and dc-ETM?

**Limitations:**

Do not see clear issues.

Post discussion: I incline to retain my positive rating.

**Strengths And Weaknesses:**

Novelty/Quality/Contributions:

The novelty of this draft is not on the techniques used; Actually, the techniques used in this draft are all not new. Using skip connection is heavily motivated by [15, 20], while using Policy gradient is also motivated by the success of previous works like [25], [26] and [27].

The authors effectively used those techniques to solve one issue — higher layers of deep PTMs/NTMs exhibit similar patterns — observed by [15]. This is the major novelty and contribution of the paper.

The authors also conducted thorough experimental study to support their claims. The authors found the proposed methods do outperform the baseline methods in terms of point log-likelihood (especially for higher-layer representations), Perplexity, topic diversity, clustering purity/NMI, diversity of generated topics (especially for higher-layer) and visual separability on topic embeddings. The authors also show the consistent benefits obtained from policy gradient based training on all the three datasets.

This work does seem to be a high-quality draft to me, though I do have some questions for the authors in the Questions section.


Clarity:

Overall, the draft is not difficult to follow, with some potentially small issues:
1. The relationship of the projection matrices of different layers are not realized in the Figure 1(c)

2. Line 177 and Equation (9) are not consistent

3. Do superscript l (line 105) and superscript (l) (line 100) indicate the same thing?


Significance:

Given good quantitative and qualitative performance, the proposed methods should be of sufficient interest to the topic modeling community.

---

> ### Author Response · Authors · 2022-08-02
> **Response to Reviewer kqfN**
>
> Thanks for your effort in reviewing this paper!
>
> **For Issue 1**
>
> Thanks for your suggestion! We provided an additional Fig. 8 in Appendix I to illustrate the detailed implementation of these two projection methods ($\alpha$ and $\beta$).
>
> **For Issue 2**
>
> Thanks for your careful reading! We have revised it and highlighted it with blue color.
>
> **For Issue 3**
>
> Yes, superscript l lacks a "()", and we have revised it in the revision.
>
> **For Question 1**
>
> Thanks for your constructive suggestion.
>
> As the implementation of two projection methods $\alpha$ and $\beta$ shown
> in Fig. 8 of Appendix I,  from the perspective of model design, dc-etm-$\alpha$ builds the straightforward connection between the observation $x_n$ and latent document representation $\theta_n^\{(l)\}$ via the projection matrix obtained by multiplying a series of topic embedding matrices, while dc-etm-$\beta$ only needs to multiply two topic embedding matrices.
>
> Thus, in our consideration, with a shorter path for gradient propagation, the short connections in dc-etm-$\beta$ can perverse more data information than those in dc-etm-$\alpha$, resulting in more informative latent document representations to achieve better model performance with less "posterior collapse''.
>
>
> **For Question 2.1**
>
> The initial motivation starts from the sequence-like generation process of dc-ETM as discussed in Section. 3.2 and recent successes of combining reinforcement learning (RL) methods with sequence generation also inspire us.
>
> However, we emphasize that the developed dc-ETM is the first one to treat the typical top-down generative process of hierarchical topic models "$ \theta_n^\{(L)\}\rightarrow \theta_n^\{(L-1)\}\rightarrow ... \rightarrow \theta_n^\{(1)\} \rightarrow x_n$" as a sequence generation task, which can be formulated as a Markov decision process and then RL-based methods can be naturally applied for model training as discussed in Section 4.2.
>
> The motivation for using Policy Gradients is that, directly training a topic model with EBLO can not update the model parameters until the whole sequence is generated, which is more likely to generate an in-balanced sequence of bad performance [31].
> Considering RL is a well-known method for learning a long sequence (Markov Decision Process) of good quality, we choose to apply the Policy Gradients, which is one of the most stable algorithms in RL, for training the developed dc-ETM.
>
> The potential reason behind the success of applying Policy Gradients for model training is that, it can decouple the target of the whole sequence generation into a series of generations of subsequence, and then the training objective at each time step will not only focus on the generation quality at the current time step, i.e., $r_t$ in Eq. (14), but also consider the future rewards collected from the following time steps $E_\pi\left[\sum_\{i=1\}^\{l-1\}\gamma^i r(s_n^\{(l-i)\}, a_n^\{(l-i)\})\right]$ in Eq. (15).  Thus, with PG-based training algorithms, we can empirically force the latent variables $\theta_n^{(l)}$ at each hidden layer to collect more data information from those at lower layers, specifically $\theta_n^\{(l-1)\}, ..., \theta_n^\{(1)\}$, and further improve the quality of these latent document representations learned by dc-ETM, leading to better model performance on downstream tasks.
>
> [31] Seqgan: Sequence generative adversarial nets with policy gradient.
>
>
>
> **For Question 2.2**
>
> Thanks for your careful reading! The reason that PG-training does not seem to benefit point log-likelihood is because that the point log-likelihood is only a sub-term of the training objective in our method, while other traditional topic models treat it as the whole objective, resulting in meaningless latent representations at higher layers. The ultimate goal of our method (training objective) is to alleviate "posterior collapse'' in deep topic models and the promising results of PG-based models shown in Table 1 and 2 demonstrate the benefits of alleviating "posterior collapse'' brought by our model design equipped with training algorithms.
>
> **For Question 2.3**
>
> Thanks for your suggestions. Due to the limited rebuttal time, we cannot finish these experiments you mentioned, but we promise that we will try to update the t-sne comparisons of dc-ETM variants in the following few days if we can.
>
> **For Question 3**
>
> Thanks. The underlying reason for this phenomenon is that traditional topic models will put almost all the pressure for generation on the first hidden layer, which makes the first hidden layer to be the most informative but the higher-layer representations meaningless.
> We note that we have provided an additional layer-wise comparison of SawETM and dc-ETM with the clustering task as shown in Appendix L.

---

> ### Author Response · Authors · 2022-08-09
> **Response to Reviewer kqfN for our promised items**
>
> Thanks again for your effort in reviewing our paper!
>
> As your awesome suggestion in **"Questions 2.3 Can the authors show qualitative comparison (e.g., t-sne plot of topic embeddings) between dc-ETM-alpha, dc-ETM-beta and dc-ETM-beta (Policy)."**, we add t-sne visualizations in **Appendix  M** for these variants' topic embeddings to investigate the effect of variant $\alpha$, $\beta$, and $policy$.

---

### Meta-Review · Area_Chair_5TN3 · 2022-08-26

**Recommendation:** Accept
**Confidence:** Less certain

**Metareview:**

This paper proposes a new hierarchical neural topic model that alleviates the posterior collapse problem of previous models. The key idea is to incorporate skip connections into the generation to alleviate the issue that higher layer representations exhibit similar patterns. The sequence-like generation procedure is formulated as a Markov decision process and learned with a policy gradient method.

Overall, most reviewers feel positively about this paper. Even though the specific techniques used are not novel (policy gradient, skip connections) their use to alleviate posterior collapse in higher layers of neural topic models seems novel. The experimental results are convincing, and the qualitative evaluation support the claim of overcoming "posterior collapse." The proposed method is potentially useful and is applicable to different types of neural topic models. Although there were some concerns in the original version (e.g. insufficient explanation on posterior collapse and why the proposed method addresses it; evaluation limited to perplexity) the authors' response addressed most of the concerns and added details in the updated version. Therefore, I recommend acceptance.

**Award:**

No

---

### Decision · Program_Chairs · 2022-09-14

Accept